# Single cell analyses reveal contrasting life strategies of the two main nitrifiers in the ocean

Katharina Kitzinger [1,2]*, Hannah K. Marchant [1]*, Laura A. Bristow[1,3], Craig W. Herbold [2], Cory C. Padilla[4], Abiel T. Kidane[1], Sten Littmann[1], Holger Daims[2,5], Petra Pjevac [2,6], Frank J. Stewart[4], Michael Wagner [2,5,6,7] & Marcel M.M. Kuypers[1]

Nitrification, the oxidation of ammonia via nitrite to nitrate, is a key process in marine nitrogen (N) cycling. Although oceanic ammonia and nitrite oxidation are balanced, ammonia-oxidizing archaea (AOA) vastly outnumber the main nitrite oxidizers, the bacterial Nitrospinae. The ecophysiological reasons for this discrepancy in abundance are unclear. Here, we compare substrate utilization and growth of Nitrospinae to AOA in the Gulf of Mexico. Based on our results, more than half of the Nitrospinae cellular N-demand is met by the organic-N compounds urea and cyanate, while AOA mainly assimilate ammonium. Nitrospinae have, under in situ conditions, around four-times higher biomass yield and five-times higher growth rates than AOA, despite their ten-fold lower abundance. Our combined results indicate that differences in mortality between Nitrospinae and AOA, rather than thermodynamics, biomass yield and cell size, determine the abundances of these main marine nitrifiers. Furthermore, there is no need to invoke yet undiscovered, abundant nitrite oxidizers to explain nitrification rates in the ocean.

[1] Max Planck Institute for Marine Microbiology, 28359 Bremen, Germany. [2] Centre for Microbiology and Environmental Systems Science, Division of Microbial Ecology, University of Vienna, 1090 Vienna, Austria. [3] Department of Biology, University of Southern Denmark, 5230 Odense, Denmark. [4] School of Biological Sciences, Georgia Institute of Technology, Atlanta, GA 30332–0230, USA. [5] The Comammox Research Platform, University of Vienna, 1090 Vienna, Austria. [6] Joint Microbiome Facility of the Medical University of Vienna and the University of Vienna, 1090 Vienna, Austria. [7] Center for Microbial Communities, Department of Chemistry and Bioscience, Aalborg University, 9220 Aalborg, Denmark. *email: kkitzing@mpi-bremen.de; hmarchan@mpi-bremen.de

Nitrification is a key process in oceanic N-cycling as it oxidizes ammonia via nitrite to nitrate, which is the main source of nitrogen for many marine primary producers. In the oceans, ammonia is mainly oxidized to nitrite by ammonia oxidizing archaea (AOA)[1,2] and the resulting nitrite is further oxidized to nitrate by nitrite oxidizing bacteria (NOB). Most inorganic fixed N (i.e. nitrate, nitrite and ammonium) in the oceans is present in the form of nitrate (99%), and <0.1% occurs in the form of nitrite, suggesting that any nitrite formed by ammonia oxidizers is immediately oxidized to nitrate[3,4]. While the discovery of the AOA[5], which comprise up to 40% of the microbial community[6], resolved the longstanding mystery of the apparently missing ammonia oxidizers[4], it raised the question as to whether there is an equally abundant, yet undiscovered nitrite oxidizer[7]. Such an organism has to date not been found and the known marine nitrite oxidizers have a 10-fold lower abundance than AOA[8–13]. The reasons for this discrepancy in abundance are poorly understood and could be due to ecophysiological differences between nitrite and ammonia oxidizers. These likely include the lower theoretical energy gain from nitrite oxidation compared to ammonia oxidation[14] and the larger cell sizes of NOB compared to AOA[5,11,15,16]. Nonetheless, it is unclear which factors keep nitrite and ammonia oxidation rates in balance due to the lack of knowledge concerning the in situ ecophysiology of marine nitrite and ammonia oxidizers. In part, this is because nitrite oxidation is rarely investigated as a standalone process in marine systems[8,9,17,18,19] and marine nitrite oxidizers are rarely quantified[9,12,20,21].

Based on the available data, marine nitrite oxidation is carried out primarily by members of the phylum Nitrospinae[8,9,11,22], and to a lesser extent by members of the genera Nitrococcus[8,23] and Nitrospira[24,25]. To date, only two Nitrospinae pure cultures are available[15,26] and both belong to the genus Nitrospina, whilst most Nitrospinae detected by cultivation-independent approaches in the marine environment belong to the candidate genus "Nitromaritima" (Nitrospinae Clade 1) and Nitrospinae Clade 2 (refs [11,22,27]) (Supplementary Fig. 1). The two cultivated Nitrospina species display relatively high growth rates, with doubling times of ~1 day[15,26]. The genome of one of the cultivated species, Nitrospina gracilis, has been sequenced, which revealed that the key enzyme for nitrite oxidation, nitrite oxidoreductase (NXR), is closely related to the NXR of Nitrospira and anammox bacteria[28]. Furthermore, Nitrospina gracilis was shown to use the reductive tricarboxylic acid cycle (TCA) cycle for autotrophic C-fixation[28].

In contrast, the ecophysiology of the more environmentally relevant Nitrospinae, "Ca. Nitromaritima" (Nitrospinae Clade 1) and Nitrospinae Clade 2, is largely uncharacterized. A recent environmental study has suggested that these Nitrospinae clades, besides being the main known nitrite oxidizers in the oceans, also play a key role in dark carbon (C) fixation, fixing as much as, or more inorganic C in the ocean than the AOA[11]. So far, however, direct comparisons of in situ C-assimilation and growth rates of NOB and AOA are lacking. Another largely unexplored facet of Nitrospinae ecophysiology is their N-assimilation strategy. Genome-based studies have shown that many environmental Nitrospinae encode the enzymes urease and cyanase (the latter is also found in the cultured N. gracilis), which allow for assimilation of the simple organic N-compounds urea and cyanate[11,22,28,29]. Direct evidence for in situ assimilation of organic N-compounds by NOB is missing. Another role of urease and cyanase could be in "reciprocal feeding", where NOB provide ammonia oxidizers with ammonia derived from the organic N-compounds and then receive the resulting nitrite[29,30]. Thus, organic N-use likely affects the distribution and activity of marine Nitrospinae and their interactions with the AOA.

Here, we determine key ecophysiological traits of Nitrospinae and compare them to those of AOA in the hypoxic shelf waters of the Gulf of Mexico (GoM). The GoM is an ideal study site to elucidate the in situ ecophysiology of these nitrite oxidizers, as it is an area characterized by high nitrite oxidation activity, which appears to be driven by Nitrospinae as the main NOB[18]. We investigate nitrite oxidation activity and growth rates of GoM Nitrospinae in comparison to GoM AOA in the same samples[31] by combining metagenomics and metatranscriptomics with stable isotope incubations and single cell techniques. Furthermore, the in situ assimilation of the dissolved organic N (DON) compounds urea and cyanate by Nitrospinae is determined, and Nitrospinae biomass yields are compared to those of the AOA. Our results show that GoM Nitrospinae and AOA display different N-utilization strategies. While Nitrospinae use mainly DON in form of urea to meet their N-requirements for assimilation, AOA predominantly assimilate ammonium. Furthermore, GoM Nitrospinae are highly efficient in converting energy to growth, and grow significantly faster than the far more abundant AOA. Our combined results indicate that in contrast to previous assumptions, the main mechanism that maintains the difference in abundance between AOA and Nitrospinae is a different mortality rate, rather than thermodynamics, biomass yield or cell size.

## Results and discussion

**Nitrite and ammonia oxidation in the Northern GoM**. Nitrite and ammonia oxidation rates were determined during an East–West sampling transect on the Louisiana Shelf of the GoM in July 2016 (Supplementary Fig. 2). Due to summertime eutrophic conditions[32], bottom waters were hypoxic at the time (<63 μM oxygen, max. water depth at the sampled stations was 18.5 m). Hypoxic bottom waters generally coincided with highest median ammonium (320 nM), urea (69 nM), cyanate (11.5 nM), nitrite (848 nM) and nitrate (2250 nM) concentrations[31] (Fig. 1a–c, Supplementary Fig. 3). These concentrations are similar to previous observations[18].

Nitrite and ammonia oxidation rates were in a similar range, with rates between 25 and 700 nM day$^{-1}$ for nitrite oxidation and 80–2500 nM day$^{-1}$ for ammonia oxidation[31] (Fig. 1d–f). Nitrite oxidation rates were in the range of the few rates that have been reported previously from the GoM[18] and other oxygen deficient waters[3,8,9,19]. There was no clear relationship between nitrite and ammonia oxidation rates in the GoM (Supplementary Fig. 4). For example, ammonia oxidation outpaced nitrite oxidation rates at Station 2, whereas at Station 3, nitrite oxidation rates were higher than ammonia oxidation rates at 12 m and 14 m depth (Fig. 1e, f). This suggests that nitrite and ammonia oxidation at individual stations and depths are not tightly linked, which is in line with previous observations in the GoM[18] and most likely can be attributed to the dynamic conditions in this region[18]. The local decoupling of nitrite and ammonia oxidation in the GOM provides a unique opportunity to study both processes independently. There was no correlation between the nitrite oxidation rates and nitrite concentration (Fig. 2a); however, there was a significant positive correlation between ammonia oxidation rates and nitrite concentrations[31] (Fig. 2b). This indicates that in the GoM, as in most of the ocean, ammonia oxidation, rather than nitrate reduction to nitrite, was the main source of nitrite[3].

**Nitrite oxidizing community; composition and abundance**. To identify the NOB responsible for nitrite oxidation in the GoM, 16S rRNA gene amplicon and deep metagenomic sequencing were performed, and in situ metatranscriptomes were obtained. The only detectable known NOB based on 16S rRNA gene

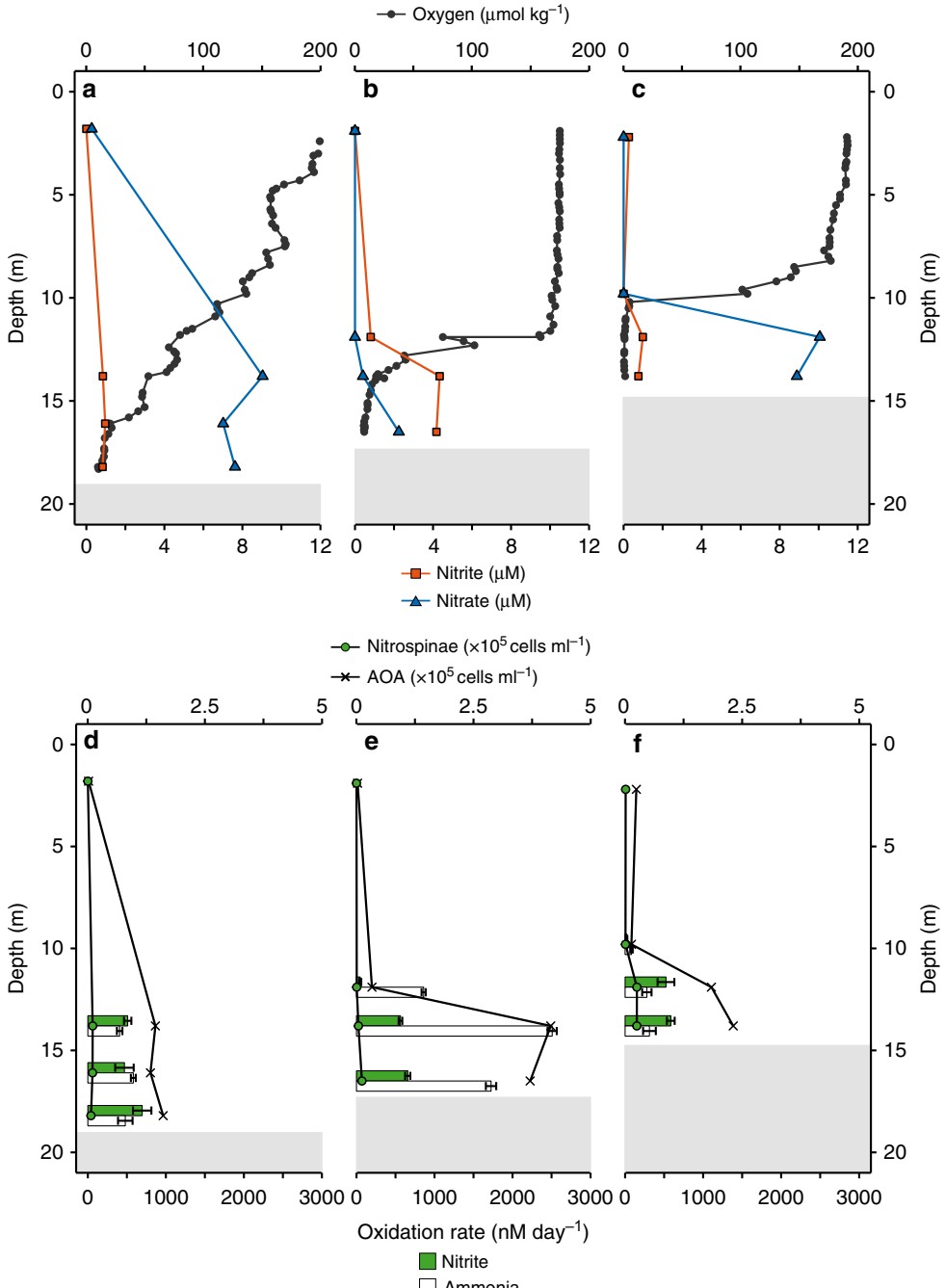

**Fig. 1 Depth profiles of nutrient concentrations, nitrite and ammonia oxidation rates, and Nitrospinae and AOA cell counts[31] at stations in the Northern GoM. a–c** In situ oxygen, nitrite and nitrate concentration profiles at Station 1 (**a**), Station 2 (**b**), and Station 3 (**c**). **d–f** Nitrite and ammonia oxidation rates and Nitrospinae and AOA CARD-FISH counts at Station 1 (**d**), Station 2 (**e**), and Station 3 (**f**). Nitrite and ammonia oxidation rates are depicted as green and white bars, respectively, and were calculated from slopes across all time points of triplicate incubations. Error bars represent standard error of the slope. Surface nitrite and nitrate concentrations (**a–c**) as well as CARD-FISH counts (**d–f**), were taken from the same station, the day before stable isotope labelling experiments were carried out. Shaded gray areas indicate sediment (max. water depth was 18.5 m) (see also Supplementary Fig. 7).

sequences in both amplicon and metagenomic datasets belonged to the phylum Nitrospinae (Fig. 3). *Nitrococcus*, another marine NOB that is frequently found in shelf areas[23], was not detected in our dataset. The metagenomes and metatranscriptomes were screened for the presence and transcription of the alpha subunit of nitrite oxidoreductase (*nxrA*), a key gene for nitrite oxidation. In line with the 16S rRNA gene results, almost all (84–98%) identified metagenomic *nxrA* fragments were affiliated with Nitrospinae (Supplementary Fig. 5). A further 2–15% of the

metagenomic read fragments mapped to *nxrA* of the NOB genus *Nitrolancea*[33]. However, only the *nxrA* genes of Nitrospinae were detected in the metatranscriptomics datasets (Supplementary Fig. 5).

Based on the retrieved metagenomic Nitrospinae 16S rRNA gene reads, several co-occurring Nitrospinae were identified: 85–94% of the metagenomic Nitrospinae 16S rRNA reads were affiliated with Nitrospinae Clade 2, 2–11% were affiliated with "*Ca.* Nitromaritima" (Nitrospinae Clade 1), and 0.1–2% were

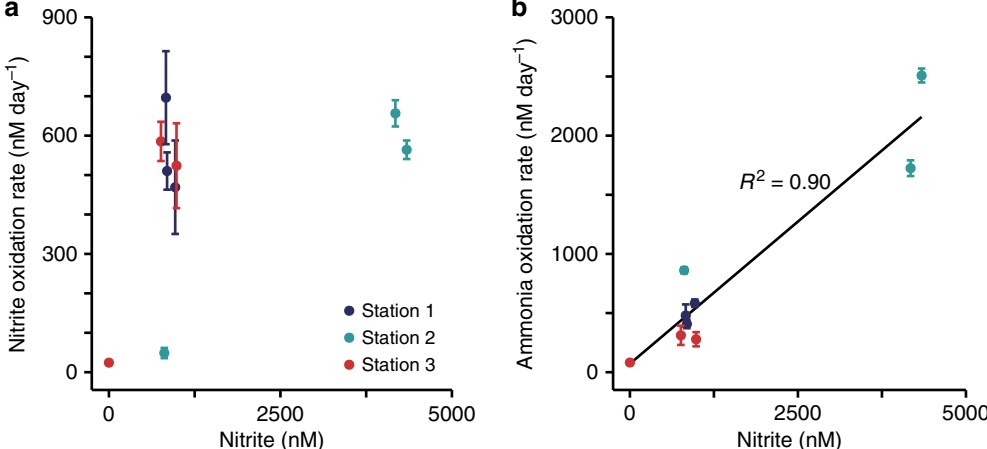

**Fig. 2 Correlations between nitrite and ammonia oxidation rates and nitrite concentrations across investigated stations. a** Correlation between nitrite oxidation rate and nitrite concentration. **b** Correlation between ammonia oxidation rate and nitrite concentration (reproduced from ref. [31]). The black line is the linear regression, $R^2$ was calculated on the basis of Pearson correlations, and was significant (two-sided $t$-test, $t = 8.002$, $DF = 7$, $P = 9.10 \times 10^{-5}$). Error bars represent standard error of the process rates calculated from slopes across all time points and replicates.

affiliated with the genus *Nitrospina*[11,22] (Fig. 3). Members of Nitrospinae Clade 2, the most abundant Nitrospinae in our dataset, are environmentally widespread and have previously been detected in metagenomes from open ocean waters, oxygen minimum zones and the seasonally anoxic Saanich inlet[11,27]. Additionally, our analyses of 16S rRNA gene sequences from global amplicon sequencing data sets (sequence read archive, SRA) revealed that phylotypes closely related (>99% identity[34]) to GoM Nitrospinae Clade 2 occur worldwide in temperate and tropical ocean waters (Supplementary Fig. 6).

To constrain absolute nitrite oxidizer cell numbers, in situ cell counts were performed by catalyzed reporter deposition fluorescence in situ hybridization (CARD-FISH) using specific probes for *Nitrococcus*[35], *Nitrospira*[36], and *Nitrobacter*[37]. We designed a new Nitrospinae-specific probe (Ntspn759), as the published Nitrospinae probes (Ntspn693 (ref. [35]) and the recently published probe Ntspn-Mod[11]) covered only a fraction of the known Nitrospinae, and did not cover all sequences in our dataset. The newly developed Ntspn759 probe targeted all of the obtained GoM Nitrospinae 16S rRNA gene sequences and 91% of the known 16S rRNA gene diversity of the family Nitrospinaceae, which contains all known Nitrospinae NOB (Supplementary Methods). The only NOB in the GoM detectable by CARD-FISH were Nitrospinae, which is in line with the observations from amplicon and metagenomic sequencing that Nitrospinae were the main NOB.

Nitrospinae were hardly detectable by CARD-FISH in the surface waters, and numbers increased with depth, reaching up to $2.8 \times 10^4$ cells ml$^{-1}$ just above the sediment. Based on CARD-FISH counts, Nitrospinae constituted at most 1% of the microbial community at all depths and stations (Fig. 1, Supplementary Fig. 7). Nitrospinae CARD-FISH counts were an order of magnitude lower than those of the only detectable ammonia oxidizers, the AOA, in the same samples (using probe Thaum726)[31,38,39] (Fig. 1, Supplementary Fig. 8a). A similar difference in abundance between these two nitrifier groups was also seen in the 16S rRNA gene amplicon dataset and the abundance of Nitrospinae and AOA metagenome assembled genomes (MAGs)[31] (Supplementary Fig. 8b, c).

The lower abundance of NOB compared to AOA in marine systems has been observed before in metagenome, amplicon, and qPCR-based studies[10,12,13,40–42]. Our results confirm this trend using CARD-FISH, a more direct quantification method that is

independent of DNA extraction and primer biases. In addition to the in situ Nitrospinae and AOA[31] counts, CARD-FISH counts were carried out at the end of the $^{15}$N and $^{13}$C incubations, which revealed that in some incubations, Nitrospinae and AOA abundances increased (up to five- and six-fold, respectively) within the incubation period of 24 h (Supplementary Data 1).

**Per cell nitrite and ammonia oxidation rates.** The per cell nitrite oxidation rate may play a key role in determining the abundance of NOB in the environment, as this rate largely determines the energy that can be gained at a single cell level. Such values have not been reported before for marine NOB, as absolute NOB cell numbers are rarely quantified at the same time as bulk nitrite oxidation rates. In fact, per cell nitrite oxidation rates have not been reported even for pure *Nitrospina* cultures. As the Nitrospinae were the only significant known NOB in the GoM, we were able to calculate per cell nitrite oxidation rates by assuming that all of the Nitrospinae detected by CARD-FISH were active (which is in line with our nanoSIMS data, see below). Average CARD-FISH cell counts between the start and the end of the incubations were combined with the bulk nitrite oxidation rates (Supplementary Data 1) to calculate per cell nitrite oxidation rates, which ranged from 21 to 106 fmol per cell per day. These rates were ~15-fold higher than the per cell ammonia oxidation rates of the AOA from the same samples (1–8 fmol-N cell$^{-1}$ day$^{-1}$)[31] (see Methods). These per cell nitrite oxidation rates are in line with those that can be estimated by combining qPCR data for Nitrospinae 16S rRNA gene abundance and bulk nitrite oxidation rates from the Eastern tropical North Pacific[9], where Nitrospinae also dominate the NOB community. Those rates ranged from 0 to 107 fmol nitrite per cell per day, assuming that Nitrospinae from the Eastern tropical North Pacific, like *N. gracilis*[28], have a single rRNA operon.

The success of NOB in oxygen deficient waters has, amongst other factors, been attributed to a high affinity for oxygen[19,43]. Our incubations were carried out at in situ oxygen concentrations, ranging from 1 to 160 μM. There was no correlation between Nitrospinae per cell nitrite oxidation rates and oxygen concentrations (Supplementary Fig. 9). This indicates that the nitrite oxidizers in the GoM were never oxygen limited, but are well adapted to low oxygen concentrations, as observed previously in other regions[19,43].

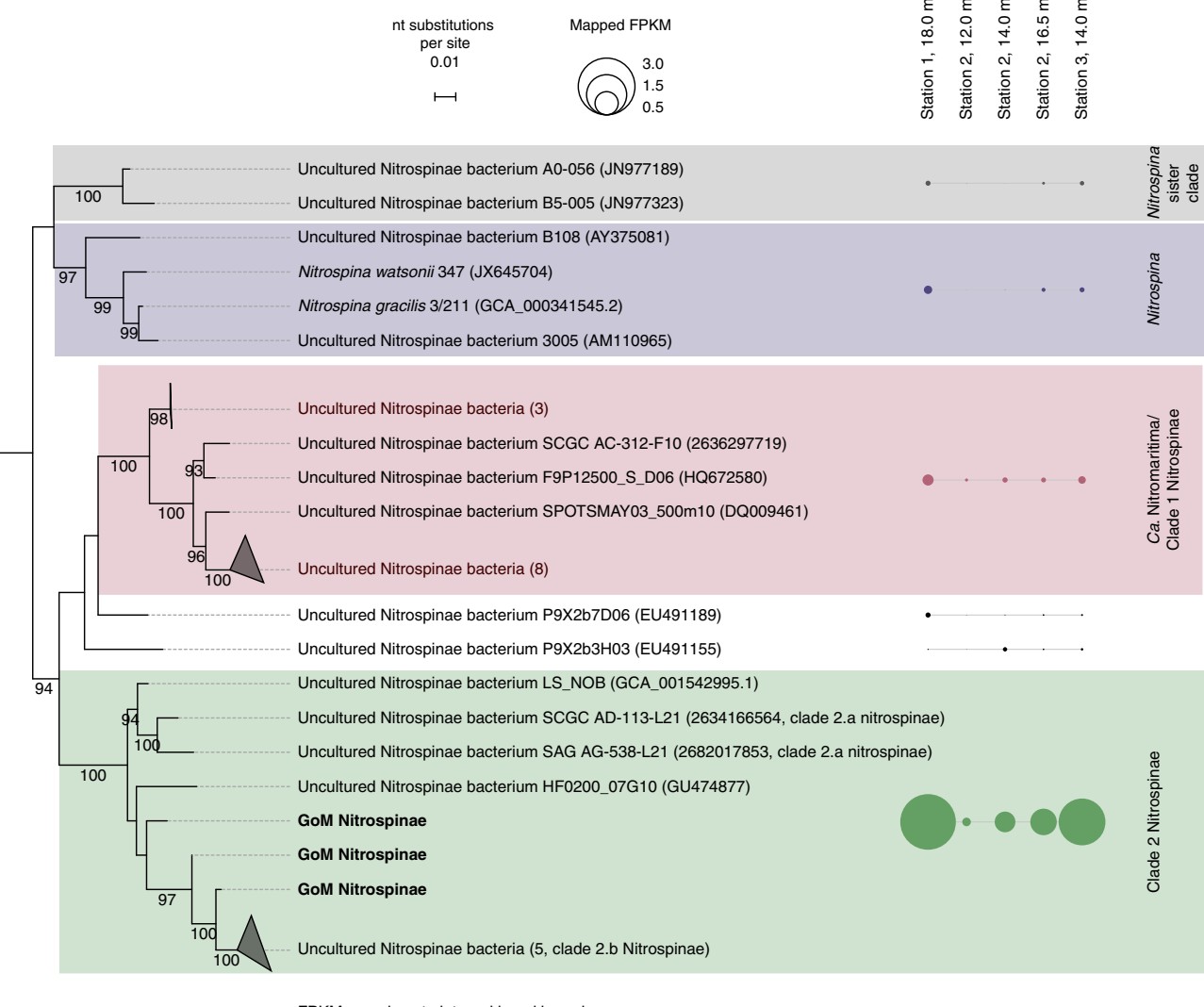

**Fig. 3 A maximum likelihood reconstruction of Nitrospinae 16S rRNA gene phylogeny.** Nitrospinae 16S rRNA gene sequences retrieved from GoM metagenomes are indicated as "GoM Nitrospinae" and printed in bold Outgroup sequences represent cultured Deltaproteobacteria. GoM metagenomic read fragments (FPKM) were mapped onto the alignment and are shown next to the respective clades as circles. FPKM mapping to internal basal nodes were grouped and are displayed separately. The scale bar represents estimated nucleotide substitutions per site, and bootstrap values >90% are displayed.

**Cellular carbon content of Nitrospinae and AOA.** Despite their low abundance, Nitrospinae have recently been estimated to be responsible for more dark carbon (C) fixation in marine systems than the highly abundant AOA[11]. This could imply that the bulk population C-content of the Nitrospinae is higher than the bulk population C-content of the AOA. Previous studies indicate that Nitrospinae cells are larger than AOA cells[5,11,15], but the differences in cell and population size have never been quantified in situ and subsequently converted to cellular or population C-content. In order to quantify the C-content of the NOB and AOA populations in the GoM, cell volumes were calculated from nanoscale secondary ion mass spectrometry (nanoSIMS) measurements. The GoM Nitrospinae were on average four-fold larger in volume than the AOA. This is in contrast to previous estimates by Pachiadaki et al.[11], who reported Nitrospinae cells to be 50-fold larger than AOA cells. However, their calculations were based on cell diameter estimates obtained from flow cytometry and assumed spherical cell shapes, whereas in the GoM and in culture, AOA resemble rods or prolate spheres and

Nitrospinae cells are curved rods[5,15,16,26]. By applying a scaling factor for C-content based on cell biovolume[44], we calculated that the GoM Nitrospinae contained approximately two times as much C per cell ($100 \pm 23$ (SD) fg-C cell$^{-1}$) as AOA ($50 \pm 16$ (SD) fg-C cell$^{-1}$, Table 1). The AOA in the GoM were visibly larger (length × width = $0.6 \pm 0.1$ (SD) × $0.4 \pm 0.1$ (SD) μm) than many cultured marine AOA[5,44–46] (length × width = $0.5–2 \times 0.15–0.26$ μm) and those normally observed in environmental studies. As such, the GoM AOA cellular C-content was higher than previously determined, ranging from 9 to 17 fg-C cell$^{-1}$ (refs [44,46–48]).

By combining the in situ Nitrospinae and AOA cell abundances and their per cell C-content, the bulk C-content of both nitrifier populations was estimated. The C-content at all investigated stations and depths ranged from 0.06 to 2.52 bulk-μg-C L$^{-1}$ for the Nitrospinae population and 0.67–20.75 bulk-μg-C L$^{-1}$ for the AOA population. Thus, the overall Nitrospinae population C-content was ~10-fold lower than that of the AOA population.

**Table 1 Parameters for estimating biomass yield by GoM AOA and Nitrospinae.**

| Parameter | GoM AOA | GoM Nitrospinae |
|---|---|---|
| Cell volume ($\mu m^3$) | 0.06 | 0.25 |
| Cell Carbon (fg-C cell$^{-1}$)[39] | 50 | 100 |
| Cell abundance (L$^{-1}$, average counts of in situ and end of incubation) | 415,000,000 | 13,200,000 |
| Bulk oxidation rate (nmol-N L$^{-1}$ day$^{-1}$) | 2508 | 564 |
| Energy gained per mol oxidized for GoM conditions based on thermodynamics (kJ mol$^{-1}$) | −262 | −65 |
| Energy gained from bulk oxidation rates based on thermodynamics (J L$^{-1}$ day$^{-1}$) | 0.658 | 0.037 |
| C-assimilation estimated from N-assimilation per population (nmol-C L$^{-1}$ day$^{-1}$) | 404 | 79 |
| Biomass yield (nmol-C J$^{-1}$, using C-assimilation estimated from N-assimilation) | 614 | 2144 |
| Measured C-assimilation from $^{13}$C-bicarbonate assimilation per population (nmol-C L$^{-1}$ day$^{-1}$) | 69 | 17 |
| Biomass yield (nmol-C J$^{-1}$, using measured C-assimilation from $^{13}$C-bicarbonate) | 104 | 464 |

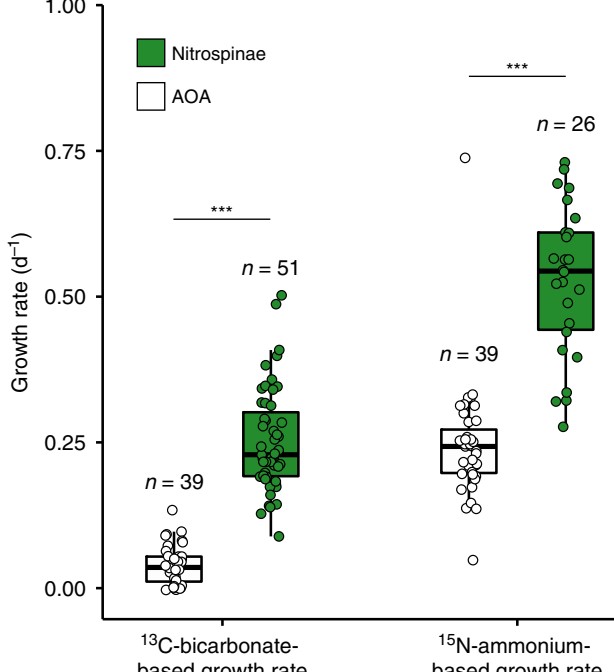

**Fig. 4 Nitrospinae and AOA growth rates calculated from $^{13}$C-bicarbonate and $^{15}$N-ammonium assimilation measured by nanoSIMS.** Nitrospinae $^{13}$C-bicarbonate assimilation rates were determined from water samples after the addition of $^{15}$N-ammonium and $^{13}$C-bicarbonate, and $^{15}$N-nitrite and $^{13}$C-bicarbonate. AOA data were acquired from the incubation with added $^{15}$N-ammonium and $^{13}$C-bicarbonate only and were taken from Kitzinger et al.[31]. Number of cells analyzed per population is indicated above each boxplot. Boxplots depict the 25–75% quantile range, with the center line depicting the median (50% quantile); whiskers encompass data points within 1.5x the interquartile range. Data of each measured cell are shown as points; horizontal position was randomized for better visibility of individual data points. Nitrospinae had significantly higher growth rates than AOA, as indicated by stars (one-sided, two-sample Wilcoxon test, $W = 1984$, $p = 4.04 \times 10^{-16}$ for growth based on $^{13}$C-bicarbonate assimilation and $W = 1464$, $p = 3.32 \times 10^{-12}$ for growth based on $^{15}$N-ammonium assimilation).

**In situ growth rates of Nitrospinae and AOA**. In situ growth rates for Nitrospinae have not been reported so far. NanoSIMS was performed on samples from Station 2, 14 m depth, which were amended with $^{13}$C-bicarbonate and $^{15}$N-ammonium (or $^{15}$N-nitrite, see Methods) to determine single cell Nitrospinae growth rates. Autotrophic growth rates from C-fixation were $0.25 \pm 0.01$ (SE) day$^{-1}$ and ammonium-based growth rates were $0.53 \pm 0.03$ (SE) day$^{-1}$ (Fig. 4), corresponding to doubling times of 2.8 and 1.3 days, respectively. The discrepancy between C- and N-based growth may be partly due to C isotope dilution by the CARD-FISH procedure[49,50]. The dilution of Nitrospinae cellular carbon by $^{12}$C-derived from the polycarbonate filters might also have affected the measured single cell $^{13}$C-uptake rates (see Methods). Additionally, the discrepancy between C- and N-based growth could indicate that the Nitrospinae use intracellular C-storage compounds to support growth or were growing mixotrophically, for which there was some evidence in the Nitrospinae MAGs (see below).

Compared to the Nitrospinae, the AOA in the GoM had significantly lower growth rates based on both $^{13}$C-bicarbonate assimilation ($0.04 \pm 0.005$ (SE) day$^{-1}$) and $^{15}$N-ammonium assimilation ($0.23 \pm 0.01$ (SE) day$^{-1}$)[31] (Fig. 4). It should be noted that the lower measured AOA autotrophic ($^{13}$C-based) growth rates may also be affected by the smaller cell size of AOA in comparison to Nitrospinae, which likely leads to a stronger C-isotope dilution effect due to $^{12}$C-derived from the polycarbonate filters (see Methods). The measured lower growth rates of AOA compared to Nitrospinae were, however, also in good agreement with substantially lower per cell oxidation rates of AOA compared to Nitrospinae.

**In situ organic N use by Nitrospinae**. Intriguingly, the ammonium-based growth rate (0.5 day$^{-1}$) of the Nitrospinae was substantially lower than that calculated from the increase in cell numbers during the incubation period, which corresponded to a growth rate of 1.2 day$^{-1}$ (0.6 days doubling time). This indicates that the Nitrospinae may have been assimilating N-sources other than ammonium. Metagenomic studies and analysis of the *N. gracilis* genome have indicated that some Nitrospinae can use the simple dissolved organic N-compounds (DON) urea and cyanate as additional N-sources[11,22,27–29]. To assess whether this is the case in the environment, single cell N-assimilation based on the incorporation of $^{15}$N-ammonium, $^{15}$N-urea, $^{15}$N-cyanate, and $^{15}$N-nitrite was determined by nanoSIMS.

All measured Nitrospinae cells were significantly enriched in $^{15}$N for all tested substrates (Fig. 5). Furthermore, the Nitrospinae assimilated significantly more $^{15}$N from all these compounds than surrounding microorganisms, including the AOA[31]. Intriguingly, ammonium and urea were assimilated equally by Nitrospinae, followed by cyanate. Nitrite assimilation by Nitrospinae was much lower compared to the other tested substrates. We calculated the growth rates of Nitrospinae and AOA from N-assimilation of all tested substrates combined (Supplementary Fig. 10). The combined N-based growth rate was 1.2 day$^{-1}$ for Nitrospinae and 0.26 day$^{-1}$ for AOA, which agrees well with cell count based growth rates of 1.2 day$^{-1}$ and 0.25 day$^{-1}$ for Nitrospinae and AOA, respectively (determined at Station 2, 14 m depth). This implies that GoM Nitrospinae and AOA could meet all of their cellular N-demand by using ammonium, urea

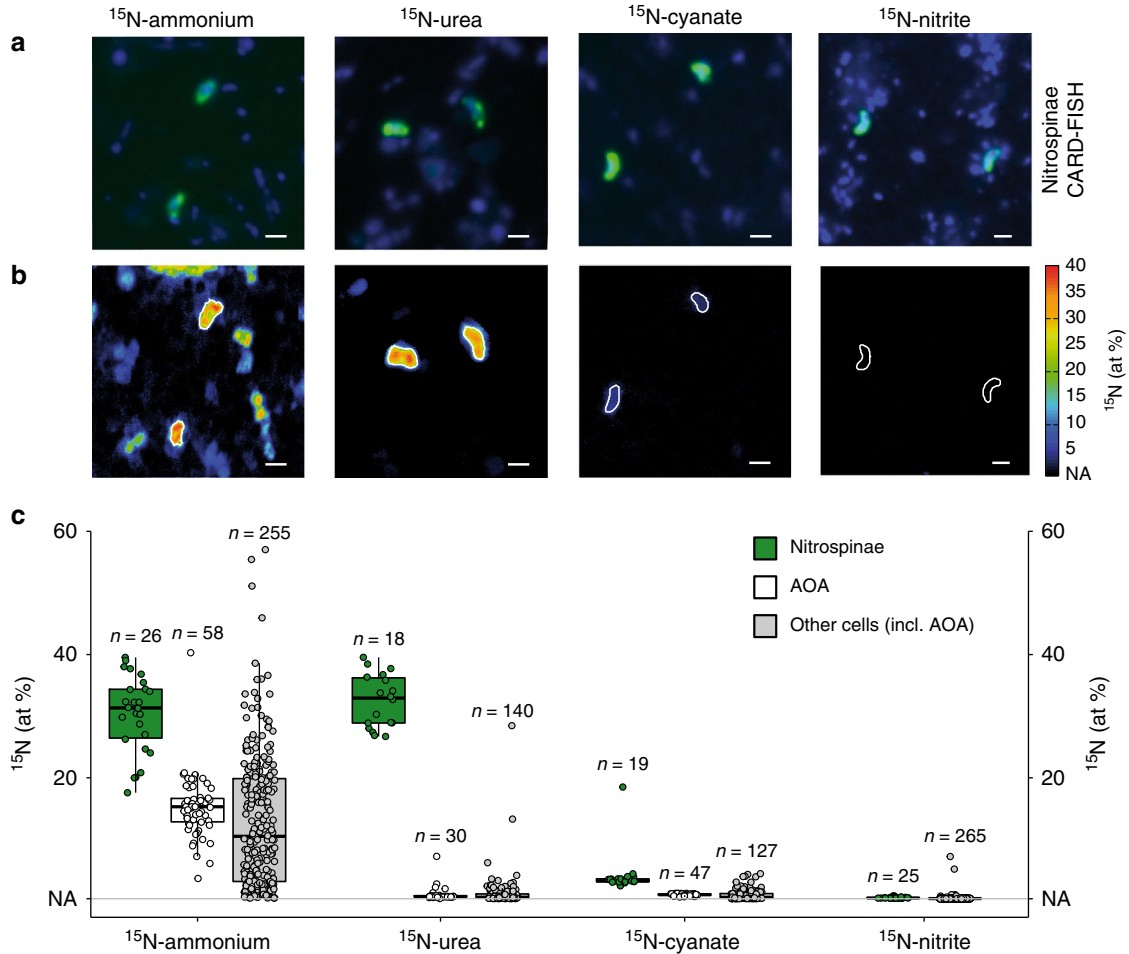

**Fig. 5 Nitrospinae single cell $^{15}$N-assimilation from ammonium, urea, cyanate and nitrite measured by nanoSIMS. a** Representative CARD-FISH images of Nitrospinae (green, stained by probe Ntspn759) and other cells (blue, stained by DAPI). **b** Corresponding nanoSIMS image of $^{15}$N at% enrichment after addition of $^{15}$N-ammonium, urea, cyanate or nitrite. Nitrospinae are marked by white outlines. Scale bar is 1 µm in all images. **c** $^{15}$N at% enrichment of Nitrospinae (green), AOA (white) and other, non-targeted cells (gray) after incubation with $^{15}$N-ammonium, $^{15}$N-urea, $^{15}$N-cyanate or $^{15}$N-nitrite. AOA data were taken from Kitzinger et al.[31] for comparison. Note that non-targeted cells depicted here also include AOA cells, as no specific AOA probe was included in the Nitrospinae nanoSIMS measurements. Number of cells analyzed per category is indicated above each boxplot. Boxplots depict the 25–75% quantile range, with the center line depicting the median (50% quantile); whiskers encompass data points within 1.5x the interquartile range. NA is the natural abundance $^{15}$N at% enrichment value (0.37%).

and cyanate. In fact, when taken together, urea, and cyanate assimilation met more than half of the Nitrospinae cellular N-demand, while AOA mainly assimilated ammonium. Utilization of DON for N-assimilation is likely a key factor for the ecological success of Nitrospinae, as it allows them to avoid competition with AOA, whom they depend on for their substrate, nitrite. The use of reduced DON (or ammonium) is also favored over nitrite because six reducing equivalents are required to reduce nitrite to ammonium before assimilation, which is metabolically costly. Thus, from an ecophysiological perspective, utilization of DON as N-source by Nitrospinae is highly advantageous.

**Nitrospinae MAG analyses.** To assess the genomic basis for DON utilization by Nitrospinae, we screened the GoM metagenomes for the presence and transcription of Nitrospinae-like cyanase and urease genes. From five deeply sequenced metagenomes, we obtained seven Nitrospinae MAGs, representing three closely related Nitrospinae population clusters (hereafter referred to as population cluster A, B, and C). Nitrospinae population cluster A made up 0.003–0.358% and population cluster B 0.008–0.152% of the metagenomic reads, compared to the lower abundance

population cluster C with 0.003–0.050% (Supplementary Table 1). All obtained MAGs were affiliated with Nitrospinae Clade 2 (Fig. 1, Supplementary Fig. 1). In line with the observed assimilation of $^{15}$N from $^{15}$N-ammonium and $^{15}$N-nitrite, the MAGs contained both ammonium and nitrite transporters, as well as assimilatory nitrite reductase genes (Supplementary Table 2). The nanoSIMS data implied that all measured Nitrospinae are capable of urea and cyanate assimilation. Accordingly, at least one MAG representative of each population cluster A, B and C contained urease and/or urea ABC-transporter genes, supporting the observed in situ assimilation of urea-derived N (Supplementary Table 2). Nitrospinae-affiliated urease genes were also transcribed in the GoM (Supplementary Fig. 11). Metagenomic read fragment abundance (FPKM) of Nitrospinae-affiliated *ureC* genes was very similar to FPKM values of Nitrospinae 16S rRNA (SSU) and *rpoB* gene abundance in all metagenome datasets (average $FPKM_{ureC}$: $FPKM_{SSU} = 1.2$, $FPKM_{ureC}$: $FPKM_{rpoB} = 1.7$), indicating that all GoM Nitrospinae encoded *ureC*. In contrast, clearly Nitrospinae-affiliated cyanase (*cynS*) genes were much less abundant in the metagenome datasets (average $FPKM_{cynS}$: $FPKM_{SSU} = 0.09$, $FPKM_{cynS}$: $FPKM_{rpoB} = 0.1$). In fact, only one of the MAGs

(population cluster B) contained the *cynS* gene (Supplementary Table 2); and its transcription was not detected in the metatranscriptomes (Supplementary Fig. 12). This contrasts with the obtained nanoSIMS data, where all measured Nitrospinae incorporated N from cyanate. The reason for this discrepancy is unknown. However, as *cynS* has previously been shown to undergo horizontal gene transfer[29,51], it is possible that GoM Nitrospinae contain additional cyanases not closely related to previously known Nitrospinae *cynS* genes[31].

In addition to urea and cyanate utilization genes, the MAGs also encoded for spermidine, amino acid and (oligo-) peptide ABC-type transporters, which may provide additional N- and C-sources for growth. The presence of a sugar transport system likely taking up sucrose, a fumarate/malate/succinate transporter, as well as many uncharacterized ABC transporter systems further indicated that the GoM Nitrospinae have a potential for mixotrophic growth (Supplementary Table 2). Mixotrophic growth of GoM Nitrospinae might contribute to the differences observed in [13]C-bicarbonate and [15]N-based growth rates and may contribute to their high measured growth rates and environmental success.

The Nitrospinae MAGs provided little evidence for alternative chemolithautotrophic energy generation pathways, which is in good agreement with recent findings from other oxygen deficient waters[27]. As in all other sequenced nitrite oxidizers, including *N. gracilis*[28], the Nitrospinae MAGs encoded a copper containing nitrite reductase (*nirK*). Furthermore, the MAG with the lowest abundance encoded a putative NiFe 3b hydrogenase, similar to the one found in the genome of *N. gracilis*[28]. Overall, the potential for known alternative energy generating pathways was low in the obtained MAGs of Nitrospinae Clade 2. However, it cannot be excluded that *Ca.* Nitromaritima (Nitrospinae Clade 1) and *Nitrospina*, which also occur in the GoM at lower abundance, and for which no MAGs were obtained, do have additional metabolic versatility.

**In situ N- and C-assimilation rates of Nitrospinae and AOA.** Single cell and population N- and C-assimilation rates were calculated for Nitrospinae and AOA using the [15]N-enrichment and their cellular N-content as calculated from their biovolumes (see Methods). Average Nitrospinae N-assimilation in fmol-N per cell per day was $0.42 \pm 0.03$ (SE) for [15]N-ammonium, $0.43 \pm 0.02$ (SE) for [15]N-urea, $0.05 \pm 0.01$ (SE) for [15]N-cyanate and $0.003 \pm 0.0004$ (SE) for [15]N-nitrite. Thus, the combined Nitrospinae N-assimilation from all [15]N-substrates together was 0.91 fmol-N per cell per day. In comparison to Nitrospinae, the single cell N-assimilation rates (in fmol-N per cell per day) of AOA were significantly lower, with $0.11 \pm 0.01$ (SE) for [15]N-ammonium, $0.005 \pm 0.001$ (SE) for [15]N-urea, $0.004 \pm 0.0002$ (SE) for [15]N-cyanate; and the combined AOA N-assimilation rate from all [15]N-substrates together was 0.12 fmol-N per cell per day.

Due to the probable bias in measured [13]C-enrichment measurements (see above), C-assimilation for both Nitrospinae and AOA was estimated from the measured [15]N-assimilation rates, following the Redfield ratio of C:N (6.6:1, see Methods). The combined Nitrospinae C-assimilation rate was 6.0 fmol-C per cell per day, compared to a much lower combined AOA C-assimilation rate of 0.76 fmol-C per cell per day. When these values were combined with the Nitrospinae and AOA cell counts, the population C-assimilation was ~80 nmol-C per liter per day for the Nitrospinae and ~400 nmol-C per liter per day for the AOA. Nitrospinae and AOA C-assimilation was also calculated from the increase in cell counts before and after incubation and their cellular C-content. The C-assimilation rate based on cell count increase was ~75 nmol-C per liter per day for the

Nitrospinae population, and ~480 nmol-C per liter per day for AOA; these values are similar to those calculated from the [15]N-tracer additions.

**Contrasting life strategies of Nitrospinae and AOA.** From a thermodynamic perspective, nitrite oxidation is a much less exergonic process than ammonia oxidation[14]. This is also the case under conditions representative for the GoM, where Gibbs free energy release is −65 kJ per mol for nitrite oxidation, compared to −262 kJ per mol for ammonia oxidation (Supplementary Table 3). Based on the measured bulk nitrite and ammonia oxidation rates at Station 2, 14 m depth (Fig. 1), nitrite oxidation provides ~0.04 Joule per liter per day, and ammonia oxidation ~0.7 Joule per liter per day in the hypoxic GoM waters. Therefore, from a purely thermodynamic perspective, AOA biomass should increase about ten times faster than that of the Nitrospinae in the GoM (Fig. 1). This, however, assumes that they have an equal biomass yield (i.e. they are fixing the same amount of C per Joule, see below), which likely is not the case[52]. The Joule energy gain was combined with the population C-assimilation rates of ~80 nmol-C per liter per day for the Nitrospinae, and ~400 nmol-C per liter per day for the AOA population (Table 1) to calculate the biomass yield for nitrite and ammonia oxidation (i.e. nmol-C fixed per Joule gained). Intriguingly, the biomass yield for the Nitrospinae population was ~2150 nmol-C per Joule, while AOA population biomass yield was only ~610 nmol-C per Joule (Table 1). This implies that Nitrospinae are ~4-fold more efficient in translating the energy gained from the oxidation of nitrite to C-assimilation than the AOA are in translating energy gained from ammonia oxidation. This is surprising considering that AOA use the HP/HB C-fixation pathway, which is suggested to be the most energy efficient aerobic autotrophic C-fixation pathway (requiring five ATP per generated pyruvate[53]). Nitrospinae employ the reverse tricarboxylic acid cycle (rTCA) for autotrophic C-fixation[28]. This pathway is highly energy efficient under anaerobic conditions (requiring two ATP per generated pyruvate) but is highly sensitive to oxygen[54]. A previous study has suggested that the Nitrospinae replace the oxygen sensitive enzymes by less oxygen sensitive versions[28]. Our results imply that at least under the low oxygen conditions in the GoM, the rTCA cycle in the Nitrospinae is also highly energy efficient.

However, additional factors likely contribute to the apparently higher biomass yield of Nitrospinae when compared to the AOA. According to the most recent metabolic models, the AOA must synthesize at least three enzymes to oxidize ammonia to nitrite[55,56]. It is noteworthy, that of the six electrons from aerobic ammonia oxidation to nitrite, only two directly contribute to energy conservation[51], while the other four are required for the reduction of molecular oxygen during the conversion of ammonia to hydroxylamine by the ammonia monooxygenase.

In comparison, the Nitrospinae have a shorter respiratory chain, oxidizing nitrite to nitrate in a single reaction, before transferring the two electrons from nitrite oxidation to oxygen. Additionally, the active site of NXR in Nitrospinae is located in the periplasm; therefore, the protons generated during nitrite oxidation might directly contribute to the proton motive force, and thus to ATP generation[28]. All of these factors, which are not captured in thermodynamic comparisons, could lead to a higher than predicted biomass yield of Nitrospinae compared to AOA. In this context, reverse electron transport, which is required for generating reducing equivalents for $CO_2$-fixation in Nitrospinae and AOA, must also be considered. This may be energetically more expensive for Nitrospinae compared to AOA, however, to date, no information is available that allows a meaningful

comparison of the actual energetic costs associated to reverse electron transport in Nitrospinae and AOA.

A further factor that could contribute to the apparent differences in biomass yield is mixotrophic growth of Nitrospinae, i.e. assimilation of organic C in addition to autotrophic C-fixation. Mixotrophy would lead to C-assimilation that requires less energy and thus the calculated biomass yield would be an overestimate, as it assumes that the measured N-assimilation is matched by autotrophic C-fixation (see Methods). Nevertheless, comparison of the directly measured $^{13}$C-bicarbonate (DIC) assimilation by Nitrospinae and AOA also indicated that the Nitrospinae had a much higher biomass yield (~465 nmol-C per Joule) than the AOA (~105 nmol-C per Joule, Table 1). In principle, the biomass yield of the Nitrospinae could also have been overestimated if they were using other electron donors in addition to nitrite, such as sulfur or hydrogen; however, little evidence for the use of alternative electron donors was found in the investigated Nitrospinae MAGs (see above).

Alternatively, rather than overestimating the biomass yield of the nitrite oxidizers, the yield of the AOA might have been underestimated if they were releasing significant amounts of dissolved organic C (DOC), as recently shown for AOA pure cultures[57]. If this occurs in the environment as well, it would have wide ranging implications for our understanding of the impact of the highly abundant AOA on C-cycling in the dark ocean.

The fact that the AOA outnumber NOB ten to one in the GoM and other marine systems despite lower AOA growth rates indicates a higher mortality rate of Nitrospinae than of AOA. This mortality could for example be due to viral lysis or zooplankton grazing. We did not perform experiments to assess the relative importance of these two controlling factors. However, both viral lysis and zooplankton grazing have previously been shown to play a major role in bacterioplankton population control[58].

Taken together, our results show that despite their lower in situ abundance, Nitrospinae in the GoM are more energy efficient, and grow faster than AOA. If our results can be extended to the rest of the ocean, no additional undiscovered NOB are required to account for the global oceanic balance between ammonia and nitrite oxidation. Furthermore, the results presented here show that Nitrospinae meet most of their cellular N-requirement by the assimilation of N from urea and cyanate, in contrast to AOA, which mainly assimilate ammonium. We hypothesize that differences in mortality, biomass yield and organic N-utilization between Nitrospinae and AOA are likely key factors regulating the abundances of these main nitrifiers in the ocean.

## Methods

**Sampling**. Sampling was undertaken on the Louisiana Shelf in the Northern Gulf of Mexico aboard the R/V Pelican, cruise PE17-02, from 23 July to 1 August 2016, on an East–West transect from 92°48′4″ W to 90°18′7″ W[31]. Briefly, seawater was sampled with 20 L Niskin bottles on a rosette equipped with a conductivity, temperature, depth (CTD), and an SBE 43 oxygen sensor. Water column nutrient profiles (ammonium, nitrite, nitrate, urea, cyanate) were measured at nine stations (surface to water-sediment interface at max. 18.5 m). Nitrite oxidation rate measurements, N- and $CO_2$-assimilation measurements, molecular and CARD-FISH analyses were carried out at three of the nine stations (Supplementary Fig. 2).

Nutrient sampling and analysis were carried out as described in Kitzinger et al.[31]. Briefly, samples for ammonium, nitrite and urea concentrations were measured onboard immediately after collection, following the procedures of Holmes et al.[59], Grasshoff et al.[60], and Mulvenna et al.[61], respectively. Samples for cyanate concentration measurements were derivatized onboard and stored frozen until analysis using high performance liquid chromatography (Dionex, ICS-3000 system coupled to a fluorescence detector, Thermo Scientific, Dionex Ultimate 3000)[62]. Samples for the determination of nitrate concentrations were stored frozen until analysis following Braman and Hendrix[63].

**N- and $CO_2$-assimilation and nitrite oxidation experiments**. Stable isotope experiments were done at three stations and three depths in and below the oxycline

as previously described[18,31]. These experiments were designed to assess ammonium, urea, cyanate and nitrite assimilation, autotrophic $CO_2$ (bicarbonate, DIC) fixation and nitrite oxidation rates. Briefly, seawater was filled into 250 ml serum bottles from Niskin bottles and allowed to overflow three times to minimize oxygen contamination. Serum bottles were then sealed bubble-free with deoxygenated rubber stoppers[64] and stored at in situ temperature (28 °C) in the dark until the beginning of the experiments (<7 h). All experimental handling took place under red light to minimize phytoplankton activity.

Tracer amendments (Supplementary Table 4) were made to triplicate serum bottles at each depth to investigate urea ($^{15}$N$^{13}$C-urea), cyanate ($^{15}$N$^{13}$C-cyanate), ammonium ($^{15}$N-NH$_4^+$), and nitrite ($^{15}$N-NO$_2^-$) assimilation and oxidation rates. All amendments were made as 5 μM additions. In the ammonium and nitrite assimilation experiments, 200 μM $^{13}$C-bicarbonate ($^{13}$C-NaHCO$_3$) was added to investigate autotrophic $CO_2$ fixation. Tracer aliquots were dissolved in sterile filtered seawater at the start of every experiment to minimize abiotic breakdown.

As described in Kitzinger et al.[31], after tracer addition, a 40 ml helium headspace was set in each serum bottle and oxygen concentrations were adjusted to match in situ conditions (Supplementary Data 1). Oxygen concentrations remained within 20% of in situ concentrations throughout the incubations, as determined by optical sensors in separate bottles (Firesting, Pyroscience). Samples were taken at the start of each experiment to determine the labeling percentage of $^{15}$N and $^{13}$C-DIC[59–62]. Thereafter, serum bottles were incubated in the dark at in situ temperature (28 °C). After 6, 12, and 24 h, 20 ml of seawater was sampled and replaced with He, sterile filtered and frozen. Serum bottle headspaces were again flushed with He, and oxygen was added to match in situ concentrations. After 24 h, the remaining seawater from triplicate incubations was combined, and 20 ml were fixed and filtered onto 0.22 μm GTTP filters for catalyzed reporter deposition fluorescence in situ hybridization (CARD-FISH) and 0.22 μm gold sputtered GTTP filters for nanoSIMS analyses (see below).

**Nitrite oxidation rate measurements**. Nitrite oxidation rates were determined from the increase in $^{15}$N-nitrate over time after the addition of $^{15}$N-nitrite. After the removal of any residual nitrite with sulfamic acid, nitrate was reduced to nitrite using spongy cadmium and subsequently converted to $N_2$ via sulfamic acid[8,65]. The resulting $N_2$ was then measured by GC-IRMS on a customized TraceGas coupled to a multicollector IsoPrime100 (Manchester, UK). Rates were calculated from the slopes of linear regressions across all time points from the triplicate serum bottles and were corrected for initial $^{15}$N-labeling percentage. Only slopes that were significantly different from 0 are reported ($p < 0.05$, one-sided student $t$-test, R v. 3.5.1)[66]. When non-significant regressions were found, rates are reported as below detection limit. For the determination and calculation of ammonia oxidation rates see Kitzinger et al.[31].

**$^{13}$C-DIC labeling percentage measurements**. $^{13}$C-DIC labeling percentages were determined from the first time point by $^{13}$C-CO$_2$/$^{12}$C-CO$_2$ measurements after sample acidification[67] using cavity ring-down spectroscopy (G2201-i coupled to a Liaison A0301, Picarro Inc., Santa Clara, USA, connected to an AutoMate Prep Device, Bushnell, USA).

**CARD-FISH counts and per cell oxidation and growth rates**. To visualize and quantify cells of the Nitrospinaceae family, a new oligonucleotide probe was designed (Supplementary Methods, Supplementary Fig. 13). For Nitrospinae and AOA quantification, seawater samples from each station and depth were fixed with 1% paraformaldehyde (final concentration, without methanol, EMS) for 12–24 h at 4 °C before filtration (<400 mbar) onto 0.22 μm GTTP filters (Millipore). Filters were stored frozen at −20 °C until analysis. Nitrospinae and AOA[31] abundances were determined by CARD-FISH according to Pernthaler et al.[68] (Supplementary Methods), using the newly designed Nitrospinae probe and probe Thaum726 for AOA[31,38,39]. Samples were additionally screened by CARD-FISH for other marine NOB of the genera Nitrospira (probe Ntspa662)[36], Nitrobacter (probe Nit3)[37] and Nitrococcus (probe Ntcoc84)[35] at the respective published formamide concentrations; published competitor probes were used for all CARD-FISH experiments.

Nitrospinae and AOA growth during the incubation time was assessed by CARD-FISH, and growth rates (Eq. (1)) and doubling times (Eq. (2)) were estimated according to:

$$GR = \ln(N_t/N_0)/t \tag{1}$$

$$DT = \ln(2)/GR \tag{2}$$

where GR is growth rate, $N_t$ the number of Nitrospinae or AOA cells at time $t$ (cell counts after incubation), $N_0$ the number of cells at time 0 (in situ cell counts), $t$ the incubation time in days and DT is doubling time in days.

Per cell Nitrospinae nitrite oxidation rates were estimated by combining measured bulk nitrite oxidation rates and Nitrospinae cell abundance, as determined by averaging Nitrospinae in situ counts and Nitrospinae counts after 24 h of incubation, as in Stieglmeyer et al.[69]. Per cell AOA ammonia oxidation rates were calculated accordingly, and therefore differ from previously reported rates which only took into account in situ AOA abundances[31].

**NanoSIMS analyses and single cell C-content and growth rates**. At the end of each incubation experiment, the content of triplicate serum bottles was combined. The seawater was filtered (<100 mbar) onto gold sputtered 0.22 µm GTTP filters (Millipore) at the end of the incubations, and fixed in 3% paraformaldehyde (in sterile filtered seawater) for 30 min at room temperature, washed twice in sterile filtered seawater and then stored at −20 °C. Before nanoSIMS analysis, Nitrospinae or AOA were targeted by CARD-FISH (without embedding filters in agarose) and all cells were stained with DAPI after which regions of interest were marked on a laser microdissection microscope (6000 B, Leica).

Single cell $^{15}$N- and $^{13}$C-assimilation from incubations with $^{15}$N-ammonium and $^{13}$C-bicarbonate, $^{15}$N-nitrite and $^{13}$C-bicarbonate, $^{15}$N$^{13}$C-urea or $^{15}$N$^{13}$C-cyanate were determined for Station 2, 14 m depth, using a nanoSIMS 50 L (CAMECA), as in Martinez-Pérez et al.[70]. Instrument precision was monitored daily on graphite planchet and regularly on caffeine standards. Due to the small size of most cells in the sample, they were pre-sputtered for only 10–20 s with a Cs$^+$ beam (~300 pA) before measurements. Measurements were carried out over a field size of $10 \times 10$ µm or $15 \times 15$ µm, with a dwelling time of 2 ms per pixel and $256 \times 256$ pixel resolution over 40 planes. The acquired data were analyzed using the Look@NanoSIMS software package[71] as in Martinez-Pérez et al.[70]. Ratios of $^{15}$N/($^{15}$N + $^{14}$N) and $^{13}$C/($^{13}$C + $^{12}$C) of Nitrospinae/AOA and non-Nitrospinae/AOA cells were used for calculation of growth rates only when the overall enrichment Poisson error across all planes of a given cell was <5%. The variability in $^{15}$N/($^{15}$N + $^{14}$N) ratios across measured Nitrospinae/AOA and non-Nitrospinae/AOA cells[31] was calculated in R v. 3.5.1 (ref. [66]) following Svedén et al.[72] (Supplementary Methods and Supplementary Fig. 14).

Single cell growth rates from nanoSIMS data were calculated as in Martinez-Pérez et al.[70], where cell $^{15}$N- and $^{13}$C-atom% excess was calculated by subtracting natural abundance $^{15}$N/($^{15}$N + $^{14}$N) and $^{13}$C/($^{13}$C + $^{12}$C) values (0.37% and 1.11%, respectively). These calculated values are considered conservative, as isotopic dilution of $^{15}$N/($^{15}$N + $^{14}$N) and $^{13}$C/($^{13}$C + $^{12}$C) ratios due to CARD-FISH was not taken into account[50,73]. AOA single cell assimilation and growth rates from these samples have previously been published[31].

The autotrophic growth rate calculations assume that all newly incorporated $^{13}$C as detected from single cell $^{13}$C/($^{13}$C + $^{12}$C) ratios is due to biomass increase. Biomass turnover due to recycling or replacing of cell components without net per cell growth, and utilization of intracellular C-storage compounds was assumed to be negligible. Nitrospinae autotrophic growth rates were measured in incubations with $^{15}$N-ammonium and $^{13}$C-bicarbonate (and an added $^{14}$N-nitrite pool), and in incubations with $^{15}$N-nitrite and $^{13}$C-bicarbonate. Nitrospinae $^{13}$C-growth rates did not differ significantly between these two incubations (two-sided, two-sample Wilcoxon test, $W = 240$, $p = 0.1113$, R v. 3.5.1)[66] and were therefore considered together. AOA $^{13}$C-based growth rates were obtained from $^{15}$N-ammonium and $^{13}$C-bicarbonate incubations only.

For estimation of the per cell C-content, cell volumes of AOA and Nitrospinae in the GoM were calculated from nanoSIMS ROI areas. For Nitrospinae, cell shapes were assumed to resemble cylinders topped by two half spheres, whereas AOA cell shapes were assumed to resemble prolate spheroids[74]. Nitrospinae and AOA cellular C-content was calculated according to Khachikyan et al.[44], and cellular N-content for both groups was calculated from C-content assuming Redfield stoichiometry (C:N = 6.625:1), as no environmental C:N ratios of AOA or Nitrospinae are available[16,44]. This in good agreement with C:N ratios published previously for cultured AOA[44] and energy dispersive spectroscopy measurements performed on *Nitrospina gracilis* (C:N = 5.9 ± 1.2 (SD), $n = 13$) according to Khachikyan et al.[44].

N-assimilation (and correspondingly C-assimilation from $^{13}$C-bicarbonate) rates were calculated by:

$$\text{N\_AssimilationRate} \left[\text{fg-N cell}^{-1}\text{d}^{-1}\right] = \left(^{15}\text{Nat\%excess}_{\text{cell}}\right)/\left(^{15}\text{Nat\%excess}_{\text{label}}\right) \times \text{fg-N}_{\text{cell}} \times 1/t \tag{3}$$

$$\text{N\_AssimilationRate} \left[\text{fmol-N cell}^{-1}\text{d}^{-1}\right] = \text{N-AssimilationRate}\left[\text{fg-N cell}^{-1}\text{d}^{-1}\right]/14 \tag{4}$$

where $^{15}$Nat%excess$_{\text{cell}}$ and $^{15}$Nat%excess$_{\text{label}}$ are $^{15}$N-atom% of a given measured cell and of the $^{15}$N-enriched seawater during the incubation after subtraction of natural abundance $^{15}$N-atom% (0.37%). fg-N$_{\text{cell}}$ is the assumed N-content per cell, and t is the incubation time in days[75].

In addition to the directly measured C-assimilation rates from $^{13}$C-bicarbonate fixation, C-assimilation rates were calculated from the measured N-assimilation rates, assuming that 6.625 mol of C is assimilated per assimilated mol of N. This was done because nanoSIMS measurements of cells filtered onto polycarbonate filters might lead to a possible dilution with $^{12}$C through edge effects between cell boundary and filter. This would affect the $^{13}$C-enrichment of all cells in the samples but likely has a larger impact on small cells like AOA.

**DNA and RNA analyses**. Samples for DNA and RNA analyses were collected from the same depths and CTD casts sampled for assimilation and oxidation rate experiments as described in Kitzinger et al.[31]. For details on nucleic acid extraction please refer to the Supplementary Methods.

**16S rRNA gene sequencing and analysis**. 16S rRNA gene diversity was assessed by amplicon sequencing, following an established pipeline[31,76,77], using barcoded primers F515 and R806 (ref. [78]). Amplicons were sequenced on the Illumina MiSeq Platform using a Reagent Kit v2 (500-cycles) and a Nano Flow Cell. Details on PCR conditions and bioinformatic analyses are described in the Supplementary Methods.

**Metagenome sequencing, assembly and binning of MAGs**. Metagenomic libraries were constructed and sequenced as in Kitzinger et al.[31] (see Supplementary Methods). Read sets were quality filtered using BBduk (BBMap v. 36.32 - Bushnell B. - sourceforge.net/projects/bbmap/), assembled using Metaspades v. 3.10.1 (ref. [79]) and binned with Metabat2 v 2.12.1 (ref. [80]) (see Supplementary Methods). Nitrospinae metagenome assembled genomes (MAGs) were identified using GTDB-Tk v. 0.2.2 (https://github.com/Ecogenomics/GtdbTk) with database release 86, which is based on the Genome Taxonomy Database[81]. To improve Nitrospinae MAG quality, the MAGs were iteratively re-assembled and re-binned (see Supplementary Methods). Metagenome sequencing statistics and information on dereplicated Nitrospinae MAGs are listed in Supplementary Tables 5 and 1, respectively.

**Metatranscriptome sequencing**. Metatranscriptomes from Station 2 were obtained as described in Kitzinger et al.[31]. To enrich for mRNA, ribosomal RNA (rRNA) was depleted from total RNA using the Ribo-Zero™ rRNA Removal Kit for bacteria (Epicentre). mRNA-enriched RNA was converted to cDNA and prepared for sequencing using the ScriptSeq™ v2 RNA-Seq Library preparation kit (Epicentre) and sequenced on an Illumina MiSeq using a 600 cycle kit. Metatranscriptomes were separated into ribosomal and non-ribosomal partitions using SortMeRNA v. 2.1 (ref. [82]). Metatranscriptome sequencing statistics are listed in Supplementary Table 6.

**Single-gene phylogenetic reconstruction**. Single-gene phylogenetic reconstruction was done as described in Kitzinger et al.[31] and is described in detail in the Supplementary Methods. Briefly, genes of interest, namely the 16S rRNA gene, and the genes encoding for nitrite oxidoreductase alpha subunit (*nxrA*), urease alpha subunit (*ureC*), cyanase (*cynS*), and bacterial RNA polymerase beta subunit (*rpoB*), respectively, were identified in metagenomic assemblies using their respective rfam and pfam HMM models. Alignments were compiled for genes (16S rRNA) and proteins (NxrA, UreC, CynS, RpoB) of interest retrieved from the GoM metagenomes and public databases. These alignments were used for phylogenetic tree calculations using IQ-TREE v. 1.6.2 (ref. [83]), using the best-fit model using ModelFinder[84] (model TNe + R3 for the 16S rRNA gene; LG + R4 for NxrA, UreC and CynS). The resulting trees were visualized using ITOL[85]. Phylogenetic trees of GoM UreC and CynS have previously been published[31], but have been recalculated using the data of the new metagenomic assembly and updated reference sequences. Accession numbers of reference sequences included in all phylogenetic analyses are given in Supplementary Data 2.

The abundances of genes of interest in metagenomic and metatranscriptomics datasets were assessed by identifying reads with BLASTX queries against the dataset assembled for phylogenetic analysis and phylogenetic placement using the evolutionary placement algorithm[86]. Read mapping is reported as fragments per kilobase per million reads (FPKM) values. FPKM values were calculated based on the number of read pairs for which one or both reads were placed into a specified location in the tree, divided by the average gene length in the reference alignment (in kb) divided by the number of total metagenomic read pairs or ribosomal-RNA free metatranscriptomic read pairs (in millions).

The percentage of *ureC*- and *cynS*-containing Nitrospinae was estimated for each metagenomic dataset as in Kitzinger et al.[31]. FPKM for urease or cyanase genes (FPKM$_{ureC/cynS}$) classified as Nitrospinae *ureC/cynS* and the FPKM for Nitrospinae *rpoB* (FPKM$_{rpoB}$) and SSU (16S rRNA genes, FPKM$_{SSU}$) genes were compared, under the assumption that *rpoB* and SSU were universally present in all Nitrospinae as single copy genes. The percentage of *ureC*-/*cynS*-positive Nitrospinae was then calculated as FPKM$_{ureC/cynS}$/FPKM$_{rpoB}$ and/or as FPKM$_{ureC/cynS}$/FPKM$_{SSU}$.

**Nitrospinae 16S rRNA gene distribution analysis**. Full-length 16S rRNA gene sequences obtained from GoM Nitrospinae Clade 2 MAGs were used to screen for the presence of closely related sequences in all publicly available Short Read Archive (SRA, www.ncbi.nlm.nih.gov/sra) datasets with IMNGS[34], using a minimum identity threshold of 99% and a minimum size of 200 nucleotides. Metadata for SRA datasets were obtained from NCBI, and latitude/longitude coordinates were plotted using the maps and ggplot2 libraries in R v. 3.5.1 (ref. [66]).

**Reporting summary**. Further information on research design is available in the Nature Research Reporting Summary linked to this article.

## Data availability

All sequence data and Nitrospinae MAGs generated in this study are deposited in NCBI under BioProject number: PRJNA397176. Metatranscriptomes are deposited under BioSample numbers SAMN07461123–SAMN07461125; 16S amplicon sequencing under SAMN07461114–SAMN07461122; metagenomes under SAMN10227777–SAMN10227781

and MAGs under SAMN12766710–SAMN12766716. Accession numbers of sequences used for tree calculations (16S rRNA gene, NxrA, UreC, CynS, and genome sequences) are given in Supplementary Data 2. CTD data, measured nutrient concentrations, process rates, Nitrospinae and AOA relative abundance based on 16S rRNA gene amplicon sequencing and CARD-FISH counts are given in Supplementary Data 1.

## Code availability

No custom code was used for analyses of amplicon sequencing data and phylogenetic analyses. Code used to automatize the MAG binning is provided by the authors upon request.

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

## Acknowledgements

We thank the captain and crew of the R/V Pelican PE17-02 cruise. We are grateful to P. Hach, G. Klockgether, D. Tienken and D. J. Parris for technical support; W. Mohr, B. Kartal, G. Lavik, N. Lehnen, A. Müller and S. Ahmerkamp for fruitful discussions. The authors thank A. Müller for providing *N. gracilis* biomass for CARD-FISH evaluations and EDS measurements. The authors thank C. Dupont for providing the *Nitrospira marina* CynS sequence. This research was funded by the Max Planck Society, the European Research Council Advanced Grant project NITRICARE 294343 (to M.W.), the Austrian Science Fund (DK plus W 1257) and the National Science Foundation grants 1558916 and 1564559 (to F.J.S.).

## Author contributions

K.K., H.K.M., M.M.M.K. L.A.B. and M.W. designed the study. K.K. and L.A.B. performed experiments, K.K. designed the CARD-FISH probe. S.L. performed EDS. S.L. and A.K. ran nanoSIMS analyses. K.K., H.K.M. and L.A.B. analyzed samples and data. C.C.P. sampled for molecular analyses, C.W.H. and C.C.P. performed molecular analyses with contribution from F.J.S., P.P. and H.D. The paper was written by K.K., H.K.M., M.M. M.K. and L.A.B. with contributions from all co-authors.

## Competing interests

The authors declare no competing interests.
