## [Peer Review File · Nature Communications]

Reviewers' comments:

Reviewer #1 (Remarks to the Author):

The manuscript by Kitzinger presents an interesting compilation of data from the Gulf of Mexico and attempts to resolve several issues in the oceanic nitrogen cycle associated with nitrification. My overall opinion is that the dataset is quite novel and well-deserving of publication, but not in its present form. There are several related issues that need to be addressed with this study:

1. The idea that nitrite oxidation must take place more rapidly than ammonia oxidation isn't new and there are many contributions in the literature that demonstrate this. Likewise, the discovery and documentation of Nitrospina (and other NOB) has largely resolved issues regarding the high abundance of AOA throughout the ocean. The overall focus of the paper is therefore not particularly novel. However, NOB are clearly critically important in the ocean, and this study provides a number of interesting pieces of information -- including new CARD-FISH probes, omic data, and esp. the nanoSIMS data. I recommend that the authors refocus on the Nitrospina data to produce a more compelling manuscript.

2. Related to this, the arguments presented by the present manuscript rest on the idea that their sampling region is generalizable to the global ocean. A key missing link to establish this would be to show that their 16S data and/or MAGs are broadly representative of Nitrospina throughout the ocean. This is hinted at in the manuscript, but no explicit analysis has been conducted. I think this will be important in order to convince many microbial oceanographers who might be skeptical that the GoM is representative.

3. Also related to point 1, the reliance on Kitzinger's earlier publication on AOA makes interpretation of the data and methods challenging. Of particular relevance are the growth rate data. More detail is needed on lines 511-515, rather than referring to the earlier publication. The AOA data in Figure 5 also need greater explanation.

4. That said, the Figure 5 data are very compelling, particularly when combined with the omic data. Do the authors have any more direct (nanoSIMS) evidence for mixotrophy in the Nitrospina? This would help support their arguments on lines 317-319 and 385-389.

5. Related to the points above, and in contrast to Figure 5, the data in Figure 4 are not as compelling. The authors use a non-parametric test to argue for statistically significant differences between the Nitrospina and AOA, but have they tested for normality, transformed, and used parametric tests (ANOVA) on the data? There is clearly a wide spread in the Nitrospina data - which probably merits discussion - and additional tests should be conducted and reported. This is obviously important for the present formulation of the manuscript, since the main conclusions rest on the idea that AOA and

Nitrospina growth rates differ. More explanation re the differences between the bicarbonate-based and ammonium-based estimates is also needed; while mixotrophy may contribute to this for the Nitrospina, and there are methodological arguments, are there other possibilities?

In sum, I think all of these are related and draw away from an interesting dataset. I would argue that the strongest parts of the study are the organic N uptake data, and corresponding omic data, as well as the interpretation of those data. Revising and refocusing on those data, while placing the other data into context, would present an interesting contribution to the literature.

Reviewer #2 (Remarks to the Author):

This manuscript seeks to provide an ecophysiological explanation for the large difference in relative abundance of ammonia-oxidizing archaea (AOA) and Nitrospinae commonly encountered in marine water columns. In particular, the authors use an elaborate and quite powerful combination of approaches, including meta-omics, stable isotope incubations, and single cell techniques, to examine the two main players responsible for nitrification in the Gulf of Mexico.

Specific comments:

Line 40: Before this, was the field really pondering where the 'missing' ammonia oxidizers were in the ocean? Might be worth citing Costa et al. (2006) Trends Microbiol paper here.

Line 102: Are the rates really 'comparable' given there are orders of magnitude difference between the two rates at Station 2?

Also, the authors state below that there was no clear relationship between the two rates (Lines 105-106).

Lines 107: But at 10m at Station 3, ammonia oxidation rate seems to be higher than nitrite oxidation rates

Line 175: What was the difference in AOA abundance pre and post-incubation?

Line 213: "The AOA in the GoM were visibly larger than cultured marine AOA ..." - This seems a little hand-wavy to me.

Could the authors provide numbers for comparison (since they did calculate cell volume)?

Their estimated C-content per AOA cell seems much higher than previously determined numbers (Line 215).

Line 228-229: If this is a concern, how are C-uptake rate measurements usually made (or supposed to be made)?

Line 308: Shouldn't these ratios be FPKM of *cynS* to SSU/*rpoB*?

Lines 311-312: Couldn't this be examined using the *cynS* tree in Supplementary Fig 11?

Lines 378-380: The authors seem to be ignoring the energetic disadvantages of having such a short respiratory chain. Are these NOB somehow getting around the need for reverse electron transport?

Line 404: References?

Line 597: The MAG accessions are missing!

Supplementary

Line 266: Wouldn't the same apply to MAG 36C, since it also has a relatively smaller binned genome size?

It is also less than 50% complete, which would make it a low-quality draft genome.

Line 149: Is '-automated1' a parameter argument for trimal?

Fig 10, 11: Would be nice to highlight the sequences binned in each of the MAGs in Table 1.

Reviewer #3 (Remarks to the Author):

On the whole, I think this is a good contribution to the understanding of marine microbial ecology, although my knowledge of the current biogeochemical literature is not strong enough to evaluate whether this is truly a novel finding. I only have a few comments, mostly on the MAG methods.

Under 'Nitrospinae MAG analyses' and also in supplementary table 2, The authors describe three clusters (A, B, and C) of Nitrospina MAGs, but I find no description of how the clusters were determined. I see that metabat was used to cluster scaffolds into MAGs, and that MAGs were refined using an iterative reassembly and rebinning approach. It's not clear from either of these steps where clusters of MAGs would have been determined.

The iterative MAG refinement is difficult to parse from the provided supplementary text. The procedure is unfamiliar to me. If it is an established technique (or based on one), please provide a reference. If it is mostly novel, an explanatory figure would be immensely helpful. Additionally, any scripts used (both R scripts and any workflow or shell scripts) should be provided to ensure reproducibility.

A final minor point: It's probably an uncontroversial statement, but there should nevertheless be a citation for the likely significance of viral lysis and grazing (line 405, near the end of the Results/Discussion)

With some clarification on the MAG clusters and inclusion of or links to refinement scripts, I would recommend this manuscript for publication. I would also strongly encourage an flow chart or citation for the MAG refinement.

Answers to Reviewers on “Single cell analyses reveal contrasting life strategies of the two main nitrifiers in the ocean”

Reviewer #1 (Remarks to the Author):

The manuscript by Kitzinger presents an interesting compilation of data from the Gulf of Mexico and attempts to resolve several issues in the oceanic nitrogen cycle associated with nitrification. My overall opinion is that the dataset is quite novel and well-deserving of publication, but not in its present form. There are several related issues that need to be addressed with this study:

Thank you for your constructive feedback. We have included additional analyses and addressed your comments, as detailed below.

1. The idea that nitrite oxidation must take place more rapidly than ammonia oxidation isn't new and there are many contributions in the literature that demonstrate this. Likewise, the discovery and documentation of Nitrospina (and other NOB) has largely resolved issues regarding the high abundance of AOA throughout the ocean. The overall focus of the paper is therefore not particularly novel. However, NOB are clearly critically important in the ocean, and this study provides a number of interesting pieces of information -- including new CARD-FISH probes, omic data, and esp. the nanoSIMS data. I recommend that the authors refocus on the Nitrospina data to produce a more compelling manuscript.

Indeed, nitrite oxidation keeps up with ammonia oxidation, and we have included a further reference for this (Ward 2008, L 41), however, in contrast to balanced process rates, the abundances of AOA and NOB are far from balanced, even when accounting for all known NOB genera (see L 44-45). The underlying mechanism that explains this offset has to date been mainly attributed to differences in cell size and/or biomass yield due to the underlying differences in thermodynamics between ammonia and nitrite oxidation (e.g. as modeled in Zakem et al. 2018). We directly link *in situ* oxidation rates to single cell growth rates, cell abundances and thus to the biomass yield of AOA and Nitrospinae. We could show that in contrast to previous assumptions, Nitrospinae are far more efficient in converting energy to growth than the AOA. The only mechanism that can maintain the difference in abundance is a different mortality rate between AOA and Nitrospinae, rather than thermodynamics or biomass yield. Therefore, we believe that our results regarding these differences in abundance are novel and should be a main focus of the manuscript.

Nevertheless, we agree that especially data on Nitrospinae is scarce to date, and have thus put a stronger focus on data on Nitrospinae in the revised manuscript.

2. Related to this, the arguments presented by the present manuscript rest on the idea that their sampling region is generalizable to the global ocean. A key missing link to establish this would be to show that their 16S data and/or MAGs are broadly representative of Nitrospina throughout the ocean. This is hinted at in the manuscript, but no explicit analysis has been conducted. I think this will be important in order to convince many microbial oceanographers who might be skeptical that the GoM is representative.

We thank the reviewer for this suggestion and have performed an analysis of GoM Nitrospinae 16S phylotype distribution in publicly available 16S rRNA gene amplicon sequence read archive (SRA) datasets. In the revised manuscript, we have included a world map showing the global distribution of close relatives to GoM Nitrospinae (L 151-154, 614-619, Supplementary Fig. 6). Nitrospinae 16S phylotypes closely related to the GoM Nitrospinae (>99% identity in 16S amplicon datasets, analyzed using the IMNGS pipeline (Lagkovardos *et al* 2016)) are indeed widely distributed in temperate and tropical regions. Therefore, we think that our findings can be extended to larger oceanic regions.

3. Also related to point 1, the reliance on Kitzinger's earlier publication on AOA makes interpretation of the data and methods challenging. Of particular relevance are the growth rate data. More detail is

needed on lines 511-515, rather than referring to the earlier publication. The AOA data in Figure 5 also need greater explanation.

We have included more information for the AOA data to enable better interpretation of the acquired data and the used methods in the revised manuscript (L 524 - 544).

4. That said, the Figure 5 data are very compelling, particularly when combined with the omic data. Do the authors have any more direct (nanoSIMS) evidence for mixotrophy in the Nitrospina? This would help support their arguments on lines 317-319 and 385-389.

Unfortunately, we have not performed additional experiments on this cruise to measure mixotrophy in Nitrospinae (or AOA). Therefore, we can only discuss the indirect evidence from the Nitrospinae genome analysis and mismatch between ^{13}C -DIC- and ^{15}N -based growth rates.

5. Related to the points above, and in contrast to Figure 5, the data in Figure 4 are not as compelling. The authors use a non-parametric test to argue for statistically significant differences between the Nitrospina and AOA, but have they tested for normality, transformed, and used parametric tests (ANOVA) on the data? There is clearly a wide spread in the Nitrospina data - which probably merits discussion - and additional tests should be conducted and reported. This is obviously important for the present formulation of the manuscript, since the main conclusions rest on the idea that AOA and Nitrospina growth rates differ.

There is indeed some variability in the measured ^{13}C -/ ^{15}N -based growth rates in both AOA and Nitrospinae. However, it should be noted that even in clonal populations single cell activity can differ significantly (e.g. Schreiber et al. 2016, Nature Microbiol). Thus, we rather find it intriguing that there is this relatively low variability in the activity of co-occurring Nitrospinae or AOA in the environment. We have tested the AOA and Nitrospinae single cell growth rate data for normal distribution and equal variances to see if parametrical tests can be used to compare the data groups. Both AOA and Nitrospinae data do not meet these required criteria and thus, we have used the non-parametrical two-sample Wilcoxon test, which shows that the groups are significantly different. To enhance the readability of Fig. 4, we changed the color scheme, depict ^{13}C -bicarbonate and ^{15}N -ammonium based growth rates of AOA and Nitrospina directly next to each-other and indicate statistically significant differences between the compared groups in the figure.

More explanation re the differences between the bicarbonate-based and ammonium-based estimates is also needed; while mixotrophy may contribute to this for the Nitrospina, and there are methodological arguments, are there other possibilities?

We have discussed the two most likely factors (methodological and mixotrophy) influencing the observed differences based on the methodology used and the acquired Nitrospinae genomic data. We don't have any evidence for additional factors that may contribute to the measured difference, although there might be more.

In sum, I think all of these are related and draw away from an interesting dataset. I would argue that the strongest parts of the study are the organic N uptake data, and corresponding omic data, as well as the interpretation of those data. Revising and refocusing on those data, while placing the other data into context, would present an interesting contribution to the literature.

Reviewer #2 (Remarks to the Author):

This manuscript seeks to provide an ecophysiological explanation for the large difference in relative abundance of ammonia-oxidizing archaea (AOA) and Nitrospinae commonly encountered in marine water

columns. In particular, the authors use an elaborate and quite powerful combination of approaches, including meta-omics, stable isotope incubations, and single cell techniques, to examine the two main players responsible for nitrification in the Gulf of Mexico.

Thank you for your positive and constructive feedback. We have addressed all comments and suggestions.

Specific comments:

Line 40: Before this, was the field really pondering where the 'missing' ammonia oxidizers were in the ocean? Might be worth citing Costa et al. (2006) Trends Microbiol paper here.

We do think that the field was pondering. Until the discovery of the AOAs, ammonia oxidizers were hardly detectable in the open ocean, while biogeochemical flux considerations indicated substantial nitrification (see also Kuypers et al., 2018 Nat Rev Microb). We have included this and the Costa et al. 2006 Trends Microbiol reference in the revised manuscript.

Line 102: Are the rates really 'comparable' given there are orders of magnitude difference between the two rates at Station 2?

Also, the authors state below that there was no clear relationship between the two rates (Lines 105-106).

We have rephrased this to "in a similar range"; we did not intend to imply that they were equal.

Lines 107: But at 10m at Station 3, ammonia oxidation rate seems to be higher than nitrite oxidation rates

We have specified this accordingly.

Line 175: What was the difference in AOA abundance pre and post-incubation?

As suggested by Reviewer 2, we have now conducted AOA cell counts post incubation and have included these results in the revised manuscript. AOA cell numbers also increased during the incubations, closely matching the calculated growth rate from our nanoSIMS measurements. We have included this in the revised manuscript (L 176-179, 190-192, and L 506-508), provided this data in Supplementary Table 1 and changed our calculations throughout the manuscript to reflect the change in cell numbers pre and post incubation.

Line 213: "The AOA in the GoM were visibly larger than cultured marine AOA ..." - This seems a little hand-wavy to me.

Could the authors provide numbers for comparison (since they did calculate cell volume)?

Their estimated C-content per AOA cell seems much higher than previously determined numbers (Line 215).

We have included the average GoM AOA cell length and width and compare them to cultured representatives in the revised manuscript (L 218-220).

Line 228-229: If this is a concern, how are C-uptake rate measurements usually made (or supposed to be made)?

The approach used here is indeed the typical approach to measure C-uptake in lacustrine or marine samples with low biomass density. Usually, C-dilution effects are not addressed in these studies.

The low biomass density in these samples requires biomass concentration by filtration, and, as CARD-FISH/nanoSIMS measurements require a flat, relatively robust filter surface, the use of carbon-based filters is the only feasible option.

In theory, measurements of cells on a C-free surface (like silica, aluminum) would alleviate this C-dilution effect, however, this is not feasible for low-biomass samples.

Line 308: Shouldn't these ratios be FPKM of *cynS* to *SSU/rpoB*?

Indeed – changed accordingly.

Lines 311-312: Couldn't this be examined using the *cynS* tree in Supplementary Fig 11?

There were many diverse *cynS* sequences present in the metagenome, and some might be from Nitrospinae. However, none of these “other” *cynS* sequences were binned into a Nitrospinae MAG, rendering an assignment to Nitrospinae impossible.

Lines 378-380: The authors seem to be ignoring the energetic disadvantages of having such a short respiratory chain. Are these NOB somehow getting around the need for reverse electron transport?

Both AOA and Nitrospinae have to rely on reverse electron transport for generating reducing equivalents for C-fixation. There are indeed energetic disadvantages of electrons entering the electron transport chain at the level of $\text{NO}_2^-/\text{NO}_3^-$ in NOB, however, it is not known how the metabolic costs of reverse electron transport between AOA and NOB differ, as it is not clear where the electrons enter the electron transport chain for reverse transport in AOA. We have addressed this in the revised manuscript in L 395-399.

Line 404: References?

We have included a reference for this statement (Suttle 2007, L 419).

Line 597: The MAG accessions are missing!

The Nitrospinae MAG accessions have been added and will be publicly available shortly.

Supplementary

Line 266: Wouldn't the same apply to MAG 36C, since it also has a relatively smaller binned genome size?

It is also less than 50% complete, which would make it a low-quality draft genome.

We have included a note on this in the table legend (Supplementary Text, L 285).

Line 149: Is '-automated1' a parameter argument for trimal?

Yes, it is. We have specified this (Supplementary Text, L 156).

Fig 10, 11: Would be nice to highlight the sequences binned in each of the MAGs in Table 1.

Done as suggested.

Reviewer #3 (Remarks to the Author):

On the whole, I think this is a good contribution to the understanding of marine microbial ecology, although my knowledge of the current biogeochemical literature is not strong enough to evaluate whether this is truly a novel finding. I only have a few comments, mostly on the MAG methods.

Thank you for the positive feedback – we have expanded the information given and references for the MAG section as suggested.

Under 'Nitrospinae MAG analyses' and also in supplementary table 2, The authors describe three clusters (A, B, and C) of Nitrospina MAGs, but I find no description of how the clusters were determined. I see that metabat was used to cluster scaffolds into MAGs, and that MAGs were refined using an iterative reassembly and rebinning approach. It's not clear from either of these steps where clusters of MAGs would have been determined.

Clusters of MAGs were derived from gANI values and represent "species-level" clusters. The main text has been modified to account for this by referring to "population clusters" and the Supplementary Text has been expanded to describe the procedure used to determine these clusters.

The iterative MAG refinement is difficult to parse from the provided supplementary text. The procedure is unfamiliar to me. If it is an established technique (or based on one), please provide a reference. If it is mostly novel, an explanatory figure would be immensely helpful. Additionally, any scripts used (both R scripts and any workflow or shell scripts) should be provided to ensure reproducibility.

We have now provided references for manuscripts that developed the iterative procedure and more clearly describe the criteria used to automate the procedure.

A final minor point: It's probably an uncontroversial statement, but there should nevertheless be a citation for the likely significance of viral lysis and grazing (line 405, near the end of the Results/Discussion)

We have included a reference for this statement.

With some clarification on the MAG clusters and inclusion of or links to refinement scripts, I would recommend this manuscript for publication. I would also strongly encourage a flow chart or citation for the MAG refinement.

REVIEWERS' COMMENTS:

Reviewer #1 (Remarks to the Author):

I reviewed an earlier version of the his manuscript and I found this version to much-improved. The authors have provided significant additional context for their results and have clarified their methods, and I have no additional detailed comments.

In particular, the authors' response to one of my comments was direct and compelling, and helped convince me of the overall direction of the paper:

"in contrast to balanced process rates, the abundances of AOA and NOB

are far from balanced, even when accounting for all known NOB genera (see L 44-45). ... We directly link in situ oxidation rates to single cell

growth rates, cell abundances and thus to the biomass yield of AOA and Nitrospinae. We could show that in contrast to previous assumptions, Nitrospinae are far more efficient in converting energy to growth

than the AOA. The only mechanism that can maintain the difference in abundance is a different mortality rate between AOA and Nitrospinae, rather than thermodynamics or biomass yield."

In fact, I would argue that this text should be used to revise the abstract and introduction. Another reviewer also highlighted how differences between AOA and NOB are presented as an awkward dichotomy in the text. I think the key point is striking a balance between the key contribution of this paper-- which is the insight into Nitrospina provided by nanoSIMS and 'omics--versus the context provided by comparison to the AOA (which are previously published data). I think this manuscript is more balanced and recommend publication given the new insights provided into the biogeochemically important Nitrospina.

Reviewer #2 (Remarks to the Author):

The authors have adequately addressed my previous comments and criticisms

Reviewer #3 (Remarks to the Author):

The authors addressed my concerns and I support the publication of paper in its updated form.

Manuscript NCOMMS-19-18806A

Single cell analyses reveal contrasting life strategies of the two main nitrifiers in the ocean

Answers to Reviewers

Reviewer #1 (Remarks to the Author):

I reviewed an earlier version of the his manuscript and I found this version to much-improved. The authors have provided significant additional context for their results and have clarified their methods, and I have no additional detailed comments.

In particular, the authors' response to one of my comments was direct and compelling, and helped convince me of the overall direction of the paper:

"in contrast to balanced process rates, the abundances of AOA and NOB are far from balanced, even when accounting for all known NOB genera (see L 44-45). ... We directly link in situ oxidation rates to single cell growth rates, cell abundances and thus to the biomass yield of AOA and Nitrospinae. We could show that in contrast to previous assumptions, Nitrospinae are far more efficient in converting energy to growth than the AOA. The only mechanism that can maintain the difference in abundance is a different mortality rate between AOA and Nitrospinae, rather than thermodynamics or biomass yield."

In fact, I would argue that this text should be used to revise the abstract and introduction. Another reviewer also highlighted how differences between AOA and NOB are presented as an awkward dichotomy in the text. I think the key point is striking a balance between the key contribution of this paper-- which is the insight into Nitrospina provided by nanoSIMS and 'omics--versus the context provided by comparison to the AOA (which are previously published data). I think this manuscript is more balanced and recommend publication given the new insights provided into the biogeochemically important Nitrospina.

Reviewer #2 (Remarks to the Author):

The authors have adequately addressed my previous comments and criticisms

Reviewer #3 (Remarks to the Author):

The authors addressed my concerns and I support the publication of paper in its updated form.

We thank all reviewers for their positive feedback. In accordance with Reviewer 1, we have included in the revised manuscript that the only mechanism that can maintain the difference in abundance is a different mortality rate between AOA and Nitrospinae, rather than thermodynamics, cell size or biomass yield (abstract L 33 – 35, introduction L60-61 and L100-104).